# Photoimmunotherapy Using Cationic and Anionic Photosensitizer-Antibody Conjugates against HIV Env-Expressing Cells

**DOI:** 10.3390/ijms21239151

**Published:** 2020-12-01

**Authors:** Mohammad Sadraeian, Calise Bahou, Edgar Ferreira da Cruz, Luíz Mário Ramos Janini, Ricardo Sobhie Diaz, Ross W. Boyle, Vijay Chudasama, Francisco Eduardo Gontijo Guimarães

**Affiliations:** 1São Carlos Institute of Physics, University of São Paulo, São Carlos, SP 13566-590, Brazil; msadraeian@usp.br; 2Department of Chemistry, University College London, London WC1H 0AJ, UK; calise.bahou.12@ucl.ac.uk; 3Laboratório de Retrovirologia, Disciplina de Microbiologia, Departamento de Microbiologia Imunologia Parasitologia, Universidade Federal de São Paulo, São Paulo, SP 04039-032, Brazil; edgar.cruz@unifesp.br (E.F.d.C.); janini@unifesp.br (L.M.R.J.); rsdiaz@catg.com.br (R.S.D.); 4Department of Chemistry, University of Hull, Cottingham Road, Hull HU6 7RX, UK; R.W.Boyle@hull.ac.uk

**Keywords:** HIV immunotherapy, photoimmunotherapy, photodynamic therapy, porphyrin, phthalocyanine, HIV-infected cell, monoclonal antibody

## Abstract

Different therapeutic strategies have been investigated to target and eliminate HIV-1-infected cells by using armed antibodies specific to viral proteins, with varying degrees of success. Herein, we propose a new strategy by combining photodynamic therapy (PDT) with HIV Env-targeted immunotherapy, and refer to it as HIV photoimmunotherapy (PIT). A human anti-gp41 antibody (7B2) was conjugated to two photosensitizers (PSs) with different charges through different linking strategies; “Click” conjugation by using an azide-bearing porphyrin attached via a disulfide bridge linker with a drug-to-antibody ratio (DAR) of exactly 4, and “Lysine” conjugation by using phthalocyanine IRDye 700DX dye with average DARs of 2.1, 3.0 and 4.4. These photo-immunoconjugates (PICs) were compared via biochemical and immunological characterizations regarding the dosimetry, solubility, and cell targeting. Photo-induced cytotoxicity of the PICs were compared using assays for apoptosis, reactive oxygen species (ROS), photo-cytotoxicity, and confocal microscopy. Targeted phototoxicity seems to be primarily dependent on the binding of PS-antibody to the HIV antigen on the cell membrane, whilst being independent of the PS type. This is the first report of the application of PIT for HIV immunotherapy by killing HIV Env-expressing cells.

## 1. Introduction

Antibody-based therapies have become important clinical tools in treating chronic diseases. Different immunotherapeutic strategies, with limited success, have been investigated to target and eliminate viral-infected cells by using armed antibodies specific to viral proteins [1]. This can be achieved by linking a drug to monoclonal antibodies (MAbs), to form immunotoxins (IT) [2], radio-labeled antibodies [3,4] or other cell-binding conjugates [5,6] capable of specifically killing infected cells, such as HIV infected cells [1,7]. Photoimmunotherapy (PIT) is a targeted photodynamic therapy (PDT) that uses photosensitizer-loaded MAbs. PIT has certain advantages over immunotoxins or RIT to eradicate infected cells [8]. In immunotherapy based on immunotoxins, a MAb is conjugated to an immunogenic toxin such as ricin [9], pulchellin [10,11] or shiga toxin [12], which can elicit an immunogenicity exemplified by the anti-toxin response [3,13]. PIT is a minimally invasive treatment, which is safer and cheaper than immunotoxins or RIT [14]. While HIV RIT is drawing the attention of researchers [4], there are no studies on the possibility of HIV treatment by PIT.

In this paper, we propose HIV immunotherapy via arming HIV monoclonal antibodies with photosensitizers (PSs). Anti-HIV immunoconjugates must be targeted to the HIV envelope spike (Env) that consists of gp160 [9,15], gp120 [16], and gp41 glycoproteins [11,16]. Visible-light activated PSs initially form the excited singlet state, which can decay by emitting fluorescence or, alternatively, it reaches the more stable excited triplet state. [17]. This triplet state can involve electron or hydrogen transfer via Type I photochemical mechanisms to yield superoxide, hydrogen peroxide, and hydroxyl radicals (electron transfer process), typical of phenothiazinium dyes [18], or Type II mechanisms (energy transfer process), typical of porphyrins [19,20], and Rose Bengal [21,22]. Reactive oxygen species (ROS) can damage cell membrane via oxidization of lipids, proteins, and nucleic acids, leading to cell necrosis where cells swell and lyse to release their intracellular content causing inflammation, or alternatively cells can undergo apoptosis [23,24]. The conventional treatment using only PS and light is called photodynamic therapy (PDT), which can destroy cells nonspecifically when exposed to light [23]. Related to this, PIT is the targeted form of conventional PDT, achieved through the conjugation of PS with MAbs targeting specific cell surface receptors [14,25]. PIT can also reduce other issues, such as dark toxicity and PS aggregation in aqueous media which reduce PDT efficacy [14]. The conjugation of a photosensitizer to antibodies can be achieved through chemical modification of the ε-amino group present on lysine residues [26] or through reactions with the thiol moiety present on cysteine residues (generated via reduction of interchain disulfide bonds). However, neither of these conjugation strategies is ideal [27,28]. Lysine modification can yield heterogeneous products with a broad distribution of drug loading, and conventional cysteine modification methods utilize interchain disulfide reduction, and result in the permanent loss of structural disulfide bonds, which has been shown to potentially impact Fc function [29] and negatively affect the stability of the antibody in vivo [30,31].

Recently, we have employed pyridazinediones (PDs) to functionally re-bridge the interchain disulfide bonds antibodies (PDs) bearing orthogonal ‘clickable’ handles into the native inter-strand disulfide bonds of trastuzumab full antibody [32,33,34,35]. By utilizing PDs harboring orthogonal ‘clickable’ handles, functional groups have been added to antibodies whilst retaining the covalent interchain linkage. This technology was then combined with mild bio-orthogonal conjugation of porphyrin photosensitizers and the conjugate targeted Her2 positive breast cancer cells [36,37].

In this study, we utilized lysine and disulfide modifications to identify an appropriate photo-immunoconjugate (PIC) for our suggested PIT. Briefly, a human anti-gp41 antibody (7B2) [38] was conjugated with two different photosensitizers, cationic porphyrin with a net charge of 3+ and anionic IR700 with a net charge of 4-. We employed two different strategies for conjugation to the antibody; lysine conjugation by using phthalocyanine IRDye700DX dye [26], and “Click” conjugation by using an azide-bearing porphyrin with a strained alkyne attached via a disulfide bridge linker [27]. The comparison between PICs is of interest with regard to the dosimetry and solubility of PICs and mechanism of in vitro cytotoxicity (Figure 1A).

## 2. Results and Discussion

### 2.1. Production and Characterization of Photoimmunoconjugates PICs

Two generations of PICs were produced; Porphyrin-7B2 PIC was produced through the use of the strain-promoted azide−alkyne cycloaddition (SPAAC) methodology in two steps: firstly, the insertion of a strained alkyne harboring pyridazinediones (PD) into native interchain disulfide bonds in humanized IgG1 MAb (7B2), then, the antibody-alkyne reacted with azide-bearing water-soluble porphyrin.

As a side by side comparison, a well-known lysine-reactive and relatively hydrophilic dye, IRDye 700DX, in different molar concentrations, was covalently conjugated with primary amine groups on the structure of 7B2 MAb using a *N*-hydroxysuccinimide (NHS) ester of the photosensitizer (Figure 1B).

Chemical characterization by UV-Vis spectrophotometry showed a PD loading of 4.3 for the rebridged antibody and a DAR of 4.1 for the final porphyrin-PIC species. IR700-PICs showed DARs of 2.1, 3.0 and 4.4, which were equal to molar ratios of 3.5, 7, and 14, before conjugation, respectively (Figure 2A). The DLS results showed that the attachment of small molecule drugs to the 7B2 MAb slightly increases the molecule’s average size (hydrodynamic radius, R_h_) and polydispersity (%Pd), indicating the presence of some high molecular weight species in this region (Figure 2A,D). 7B2-porphyrin showed the highest %Pd and R_h_ in the main peak. The zeta potential of 7B2 antibody changed very slightly from +0.05 mV towards +0.01 and −0.04 for cationic porphyrin and anionic IR-700 DAR4, respectively (Figure 2A). The binding of the 7B2 based-PICs to gp41 loop peptide antigen was examined within the conjugation. ELISA results showed that an increase of IR700-antibody-ratio decreases the binding ability, as 7B2-IR700 DAR4 showed the lowest binding to the gp41. In contrast, the conjugation of porphyrin with 7B2 MAb did not significantly affect the immunologic specificity of the antibody (Figure 2B). Results of microcapillary electrophoresis of non-reduced antibody (H and L chains) and PICs demonstrated the molecular weights (MWs) with 151.4, 166.1 and 166.9 kDa for naked antibody, rebridged antibody and clicked porphyrin-antibody. The increase in the MWs of IR700-PICs were in agreement of determined DARs, with 154.1, 154.7 and 156.8 kDa for DARs 2, 3 and 4, respectively (Figure 2A,C). 

In the presence of reducing agent (TCEP), the light and heavy chains of either 7B2 or 7B2-IR700 were observed, due to the reduction of all available interchain disulfide bridges. The reduced 7B2-porphyrin did not show the separation of chains, confirming the accuracy of click chemistry with DAR4 (Figure 2C).

In total, the characterization of PICs showed that the shape, solubility and binding ability of IR700-PICs was affected by increasing the drug-to-antibody ratio (DAR), while they were preserved for water-soluble homogeneous porphyrin-7B2 with a constant DAR of 4, due to the retention of rigid structural bridges on the antibody. 

### 2.2. Flow Cytometry Analysis

By using a Cary Eclipse UV-Vis-NIR spectroscopy, we analyzed the excitation and emission spectra before flow cytometry study without using a secondary antibody. We showed the excitation of porphyrin by blue laser at 488 nm would not interfere with the emission intensity. The excitation by violet laser at 405 nm caused red emission at 725 nm with 1.163 (a. u.) intensity, while the maximum red fluorescent intensity (3.779 a. u.) can be observed by the excitation of Soret band of porphyrin at 432 nm. The results were in agreement with the theoretical analysis of UV-Vis spectra (Appendix A). 

Specific cell binding to native Env was examined by flow cytometry. Without using a secondary antibody, the red fluorescent emission from IR700-PICs and porphyrin-PIC were directly detected on the Env-transfected cells, by filter Qdot 705 and filter Qdot 655, respectively (Figure 3A). By using FITC-secondary antibody, PICs showed equal binding ability into Env-transfected cells, while 7B2-IR700 DAR4 showed somewhat unspecific binding into the control 293T cells (Figure 3B). Not surprisingly, the presence of soluble CD4 (sCD4) showed a synergic effect on the binding ability of porphyrin-7B2, in comparison to the absence of sCD4 (Figure 3C). This result is in agreement with our previous studies on immunotoxins [11,16], as we showed sCD4 enhances epitope exposure.

Interestingly, IR700-PICs with different DARs (Figure 2A) and different binding abilities (Figure 2B) showed equal fluorescence intensity, not only with the direct red emission by filter Qdot 705 (Figure 3A), also with indirect emission by FITC-secondary antibody (Figure 3B). This can be due to the quenching of chromophores, as some that are too close to each other will prevent the chromophores from being excited by laser light and then bleaching [39]. There is no linear relationship between the number of chromophores per antibody and fluorescence intensity [40], while there is a correlation between the trends in fluorescence quenching and efficacy of heterogeneous IR700-PICs [41]. We assume this quenching would not happen in homogeneous porphyrin-PIC with disulfide modification. 

### 2.3. ROS Generation

In cellular studies, to have an equal irradiation for both porphyrin-PIC and IR700-PIC, for a side by side comparison, a custom-made LED device with a broad spectrum of light (380−780 nm) was utilized, as the blue spectra can excite the Soret band of porphyrin at 432 nm, meanwhile, red spectra excites IR700-7B2 at 689 nm (Figure 4A). 

To detect the ROS generation, the incubated cells by PICs and H2 DCF-DA fluorescent probe were irradiated with 50 J/cm^2^. H2DCF-DA fluorescent probe was applied to detect ROS production in PIT-treated cells by using flow cytometry. The Env-transfected cells incubated with PICs exhibited a significant increase in ROS levels, whereas control 293T cells did not show ROS production (Figure 4B). In parallel, in the positive control group, we observed ROS production in 293T cells which were treated with 80 µM H_2_O_2_ for 24 h in the dark (Figure 4C). The irradiated cells, with either no incubation or incubated with 7B2 antibody, showed an increase in the fluorescence intensity in comparison to the unstained cells in darkness. 

### 2.4. Apoptosis Assay

For apoptosis assays, the Env-transfected cells were incubated by PICs for 4 h, irradiated, then Annexin V - FITC/PI dye added before flow cytometry analysis. No features related to early stages of apoptosis, such as extracellular exposure of phosphatidylserine, was observed in response to PICs (Figure 4D, purple color in the lower right subset). Both porphyrin-PIC and IR700-PIC showed cell death due to late apoptosis (FITC emission) after 4 h incubation, and signs of necrosis (PI emission). Control 7B2 antibody showed neither cell death nor signs of apoptosis (Figure 4D). The non-irradiated cells did not show sign of apoptosis.

### 2.5. Photo-Cytotoxicity Assay

Env-transfected 293 T cells (293 T/92UG) and control 293T cells were incubated with a serial dilution of PICs starting at 500 nM in the presence of sCD4 (anti-gp120) for 4 h then were assayed by photoimmunotherapy (PIT) with 20 J/cm^2^. IR700-PIC with a DAR of 3 appeared more cytotoxic than porphyrin-PIC with a constant DAR of 4. PICs showed neither dark cytotoxicity on transfected cells nor nonspecific killing on 293T control cells. The cell growth inhibition was observed due to the irradiation on the cells without PIC treatment (green stars in Figure 5A).

As a prospect of this approach in an in vivo study, the doses of irradiation can be increased to maximum energy fluence of 100 J/cm^2^, depending on the selected wavelength, however, visible light irradiation is unlikely to be toxic [42]. Optical microscopic observations before and after PIT demonstrated the bleb formation as signs of necrotic cell death. Cells PIT-treated with IR700-7B2 showed not only rapid bleb formation but also cell debris (Appendix A).

To study how the combination of both PICs may affect the cytotoxicity, an equal molarity of porphyrin-PIC was mixed with IR700-PIC. Interestingly, this combination showed a significant increase in the cytotoxicity in compare to the average cytotoxicity of both PICs, due to the synergic effect of two PICs with two different mechanism (Figure 5A,B).

Furthermore, the photo-cytotoxicity of porphyrin-7B2 increased twice by increasing the time of incubation from 1 h to 4 h, indicating the internalization-dependency of porphyrin-mAb. In contrast, the photo-cytotoxicity of IR700-7B2 did not change significantly on increasing the time of incubation at 37 °C (Figure 5C). 

### 2.6. Confocal Microscopy Study

The cells were treated with IR700-7B2 or 7B2-porphyrin for 1 h, then washed and irradiated by 2-photons, 800 nm, 2.2 mW/cm. Red fluorescence emission from IR700-7B2 was directly detected (Figure 6A and Appendix A). As 7B2-porphyrin did not show strong red fluorescent emission, FITC anti-human IgG secondary antibody was applied to detect PIC indirectly (Figure 6B and Appendix A). After 10 min of adding porphyrin-7B2, the cellular swelling and bleb formation were observed as signs of necrotic cell death. The same necrotic signs were observed by adding IR700-7B2, but in 5 min of irradiation. After 30 min irradiation, Acridine Orange and Ethidium Bromide (AO/EB) were added as Live-dead dyes. Interestingly, the irradiated and non-irradiated regions were clearly distinguishable in red and green colors, respectively, confirming no dark toxicity by PICs (Figure 6C).

In conclusion, we compared the mechanism of two photosensitizers in the formation of PICs regarding the induction of cell death in HIV Env-transfected cells. We also addressed some of the main issues associated with conventional PDT including low selectivity, controlled drug dosimetry, dark toxicity and low water solubility of photosensitizer. Finally, we studied the possibility of PIT application to directly killing HIV-1 Env-expressing cells. Targeted phototoxicity seems to be primarily dependent on the binding of PS-antibody to the antigen on the cell membrane or HIV Envelope, whilst being independent of the PS type. Considering the fundamental restriction of light penetration, we suggest optimizing the excitation of PSs in deep tissue that may give us a highly flexible theranostic platform for HIV researches. This inevitably issue dictates the use of high-power irradiation [42]. The other alternatives for light delivery to the PICs in deep tissue could be the application of cellular-scale inorganic light-emitting diode (µ-ILED) arrays [43], or upconversion nanoparticles (UCNPs) in the field of PIT [44]. 

Approaches that focus on the combination of the PIT strategy with “shock and kill” technique could be a cornerstone of future research efforts towards an HIV cure [45]. The photo-cytotoxicity on the cell membrane triggered by the presence of HIV proteins can be an interesting to be proven strategy aiming the elimination of HIV latent reservoir in vivo. Previously, we observed different levels of surface HIV Env on the transfected and infected cells which may reflect differences due to laboratory-adapted versus clinical isolates [11]. However, it is still unclear how these discrepancies may relate to the expressed levels of HIV Env on the surface of HIV-infected cells, especially in the context of HIV-1 latency. We therefore recognize that further experimental killing approach on HIV-1 latently infected cells would add fundamental knowledge to the proposed PIT strategy. To the best of our knowledge, this is the first report to demonstrate the utility of photoimmunotherapy for HIV Env-expressing cells. The results of this strategy can potentially translate to viral photo-immunotherapy against other enveloped viruses.

## 3. Materials and Methods

### 3.1. Consumable Materials

#### 3.1.1. Chemical Reagents

All reagents are from ThermoFisher Scientific (Waltham, MA, USA) unless a statement to the contrary is made. 

#### 3.1.2. Cell Lines

293T/92UG cell lines stably express clade A clinical isolate 92UG037.8 gp160 as native trimers of HIV gp120/gp41. In this paper, Env-transfected cells refer to 293T/92UG cell lines [46], and HEK 293T cells refer to non-transfected control cells. The transfected and non-transfected 293T cells were incubated at 37 °C in 5% CO_2_ in DMEM medium with 10% fetal calf serum (Gibco Invitrogen, Grand Island, NY, USA). 

#### 3.1.3. Antibodies

MAb 7B2 (GenBank accession numbers JX188438 and JX188439) is an Anti-gp41 hIgG1 binding at amino acids 598–604 (CSGKLIC) in the helix-loop-helix region [38,47]. HY (Genbank accession numbers JX188440 and JX188441), is hIgG1/kappa, an affinity matured version of the anti-CD4 binding site Ab b12 [48]. MAbs were purified from supernatant media of hybridoma after passing Protein A agarose beads (Invitrogen, Carlsbad, CA, USA). Two types of soluble CD4 were utilized to study CD4-mediated effects [47]; CD4-IgG2 is a hIgG2 containing a tetrameric fusion protein in which the Fv portions of both H and L chains have been substituted by the V1 and V2 domains of hCD4. CD4-183 is a soluble fragment of human CD4 that binds the HIV gp120. Goat anti-human IgG (Invitrogen) was conjugated to either fluorescein isothiocyanate (FITC) or alkaline phosphatase (AP).

### 3.2. Porphyrin-Antibody Conjugation by Click Chemistry

Conjugation between azide porphyrin and mAb 7B2 was carried out in two steps: antibody functionalization, then click chemistry conjugation. The process is a modification of our protocol described previously [35,49,50] and described in detail in the Appendix A.

### 3.3. Conjugation and Optimization of IR700-Antibody by Lysine Modification

IRDye 700DX (IR700, LI-COR Biosciences, Lincoln, NE, USA) was covalently linked to purified MAb 7B2 through an *N*-hydroxysuccinimide reactive group, according to the manufacturer’s instructions [26]. The process of conjugation and optimization are described in detail in the Appendix A.

### 3.4. UV-Vis Spectroscopy

UV-Vis spectroscopy was used to measure protein concentrations and photosensitizer-antibody ratios (DAR), firstly by using Nanodrop 1000 UV-Visible spectrophotometer (ThermoFisher Scientific) and then a Cary 100 Bio UV-Visible spectrophotometer (Varian, CA, USA) operating at 21 °C. Blank was the sample buffer to correct base line with extinction coefficients; ε_280_ = 250,440 M^−1^ cm^−1^ for antibodies, ε_335_ = 9100 M^−1^ cm^−1^ for pyridazinedione scaffolds, ε_422_ = 165,175 M^−1^ cm^−1^ for porphyrin and ε_689_ = 165,000 M^−1^ cm^−1^ for IRDye 700DX. A Correction Factor at 280 nm of 0.25 (at A_335_) was applied for pyridazinedione scaffolds, as described elsewhere [36]. 

### 3.5. Electrophoresis

Molecular size, purity and accuracy of the conjugation of products were determined using Non/reducing glycine-SDS-PAGE and then confirmed by microcapillary electrophoresis in the presence or absence of TCEP·HCl (6.0 μL, 20 mM in d.d water, 12 eq.), as a reductive agent (Agilent Bioanalyzer, GE Healthcare, Piscataway, NJ, USA), following standard lab procedures.

### 3.6. ELISA

Purified MAbs and photo-immunoconjugates (PICs) were analyzed for antigen-binding specificity and titration by using ELISA, in wells coated with gp41 antigen (1 μg/mL), as described elsewhere [16]. The gp41 antigen is a linear peptide sequence [LGIWGCSGKLICTT] representing the epitope of MAb 7B2. AP-conjugated secondary antibody (goat anti-human IgG) (Zymed Laboratories, South San Francisco, CA, USA) were used to detect the primary antibodies binding to the antigen. Data are represented as optical density at 405 nm and signify means of triplicate values with three independent experiments.

### 3.7. Dynamic Light Scattering (DLS) and Zeta Potential

Hydrodynamic radii, electrophoretic mobility, zeta potential, and polydispersity of naked antibody and PICs were measured. Samples with 70 μL volume at 1 mg/mL in UV-transparent 96-well plates were measured by using a DLS Wyatt Möbius (Wyatt Technologies, Dernbach, Germany) with incident light at 532 nm, at an angle of 163.5°. Samples were equilibrated at 25 ± 0.1 °C for 600 s before the measurements, with the constant temperature during the experiments. All samples represent triplicate values with 10 acquisitions and a 5 s acquisition time. The change in cumulant fitted hydrodynamic radius in nanometers was tracked throughout the storage period. Results were analyzed using the Dynamics software version 7.1.7 (Wyatt Technologies, Santa Barbara, CA, USA). By using an Agilent Technologies Cary 5000 Cary Series UV-VIS-NIR spectrofluorimeter, we showed the incident laser beam at 532 nm is not within the fluorescent sample’s (IR700-7B2 and porphyrin-7B2) band of excitation; as the analysis would not be disrupted or tainted.

### 3.8. Direct and Indirect Immunofluorescence Using Flow Cytometry 

Direct and indirect immunofluorescence were applied using flow cytometry to analyze the binding of photo-immunoconjugates (PICs) to Env-transfected 293T cells. 8 × 10^4^ cells were incubated with PBS/BSA/0.01% sodium azide (PBA) for 30 min in RT, and then PICs added to reach the final concentrations of 10 µg/mL in the presence of 5 µg/mL soluble CD4. Cells were incubated with PICs for 1 h at room temperature. For indirect assay, half of the samples were washed, and then stained with FITC conjugated goat anti-human IgG secondary antibody (2 μg/mL) for 1 h. All the samples, for direct and indirect assays, were washed twice and fixed in 100 μL of 2% paraformaldehyde. After 4 h, 150 μL of PBS was added to the samples. Cells with 10,000 events were evaluated on BD LSRFortessa (Becton-Dickson, Mountain View, CA, USA), analyzed by Flow-Jo software version 7.5 (Tree Star Inc., Ashland, OR, USA). Mean fluorescence of the gated cell population labelled with immunoconjugate was calculated in relation to the mean fluorescence of cells labelled with MAb 7B2. 

Direct immunofluorescence without adding FITC conjugated secondary antibody was applied to analyze the binding of PICs to Env-transfected 293T cells. For IR700-PICs, the cells were excited by either violet laser at 405 nm or red laser at 640 nm, and the red fluorescence emission was detected by 710 ± 50 band-pass filter (Qdot 705). For porphyrin-PICs, the cells were excited by violet laser at 405 nm, and the red fluorescence emission was detected by 660 ± 20 band-pass filter (Qdot 655).

### 3.9. ROS Detection 

ROS generation in the cell lines was detected with the fluorescent probe H2DCF-DA (2′,7′-dichlorodihydrofluorescein diacetate). 2 × 10^5^ cells were incubated with 1 µM of native antibody, porphyrin-PIC, or IR700-PIC in 96 well plates in phenol-free DMEM for 45 min. The cells were washed and incubated with 1 μM H2DCF-DA solution prepared in PBS, and washed again and incubated with phenol-free DMEM before irradiation with 50 J/cm^2^. A mirror dark-plate, submitted to the same procedure, except for light exposure, contained the dark control groups. The cells incubated with 40 mM H_2_O_2_, without irradiation, were used as positive control. Fluorescence intensity (Ex/Em = 485/535 nm) was measured with the flow cytometry, with the procedure as mentioned above.

### 3.10. Apoptosis/Cell Death Assay

Annexin V-FITC/PI cell staining was performed following 6 h of standard photoimmunotherapy. Briefly, cells were incubated with Annexin V binding buffer (10 mM HEPES, pH 7.4; 2.5 mM CaCl_2_; 140 mM NaCl) and stained with recombinant annexin V conjugated to green-fluorescent FITC dye (ThermoFisher Scientific, San Jose, CA, USA) for 15 min. After three washes, cells were incubated with 2 μg/mL of propidium iodide (PI) (Life Technologies, Carlsbad, CA, USA) for 15 min. The mean fluorescence intensity was measured for 10,000 events. The data were collected using linear amplification for two optical detectors, FSC and SSC, and log amplification for related filters, and analyzed by associated software as mentioned above.

### 3.11. Photo-Cytotoxicity Assay

Naked 7B2 antibody, IR700-7B2, porphyrin-7B2 and a mixture of both PICs with equal molarity were diluted in the phenol-free DMEM medium, but without FCS, to reach a range of five final concentrations of 500 nM, 250 nM, 125 nM, 62.5 nM and 31.2 nM into 5 wells of a 96-well plate. HIV Env-transfected cells and 293T cells were adjusted to a concentration of 1 × 10^6^ cells/mL. Soluble CD4 was added to each well in a final concentration of 500 ng/mL. The plate incubated for 4 h at 37 °C and 5% CO_2_. Identical plates were kept protected from light for the same period to obtain the proper dark control groups. Afterwards, the cells were washed with a 4X excess of phenol-free serum-free DMEM to remove any unbound PICs. The plates were irradiated with 20 J/ cm^2^ of light (380−780 nm) by a homemade LED array (30 mW/cm^2^) in two equal doses separated by 10 min, as described elsewhere [51,52]. After irradiation, 10 μL of FCS was added to each well and the plates were kept in the incubator for 3 days. The cell growth in each group was also monitored and imaged under a microscope (IX81, Olympus America, Center Valley, PA, USA) at each time point, before and after irradiation. MTS/PMS substrate (Promega, Madison, WI, USA) was added to cultures and absorbance at 490 nm was recorded at 1–4 h of incubation. Results represent the mean and SEM of triplicate samples for three independent experiments, and plotted as A_490_ with a subtraction from background. Data are represented as % of cell viability versus PIC concentration. 

### 3.12. Live Imaging by Two-Photon Confocal Microscopy

Live cell imaging was performed using an inverted LSM 780 multiphoton laser scanning confocal microscope (Zeiss, Jena, Germany) equipped with a Chameleon laser (Ti:sapphire, Coherent, Santa Clara, CA, USA) as a source for two-photons (2P) excitation at 800 nm. The images were obtained by the average of 2 scans and no appreciate variation was observed. The spatial resolution was approximately 350 nm considering the numerical aperture and the excitation wavelength, as described previously [53]. The occurrence of cell death (irradiated region) was compared with live cells (non-irradiated region) by acridine orange-ethidium bromide (AO/EB) double-staining. Detailed conditions and settings are described in the Appendix A.

### 3.13. Statistical Analyses

Statistical analyses were performed with GraphPad Prism version 8.0 (GraphPad Software, San Diego, CA, USA). Data are shown as mean and standard error of mean (SEM) of the indicated number of replicate values. If no error bar appears signifying that the error bars are smaller than, and obscured by, the symbol. The unpaired two-tailed Student’s t-test was applied for statistical comparison, unless specifically stated otherwise.

### 3.14. Data Availability

All data generated or analyzed during this research are included in this published article and its Appendix A. The detailed method of live microscopy with extra videos can be found at https://docs.google.com/forms/d/e/1FAIpQLSdOC3q1uWNIpVj_iPcz5G8rLrQRjuffNowE2U-DcUrKI3CAw/viewform?usp=sf_link.Password:USP4030.

## Figures and Tables

**Figure 1 ijms-21-09151-f001:**
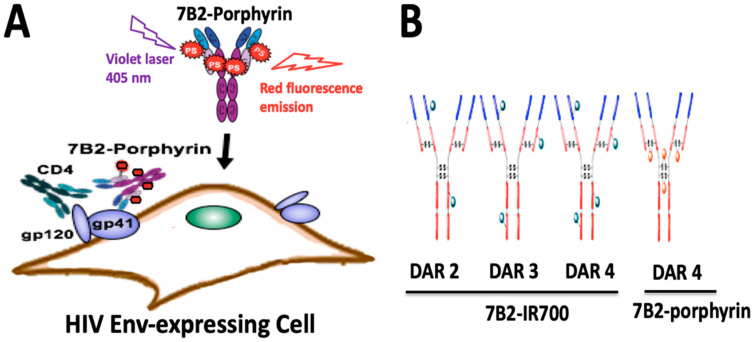
Structure and function of photo-immunoconjugates (PICs). (**A**) Anti-gp41 monoclonal antibody (7B2 MAb) conjugated with 4 porphyrins can specifically bind to the HIV infected cells expressing HIV Env (gp41 and gp120). The presence of CD4 may increase the binding ability of 7B2-porphyrin to the gp41 on the cell membrane. Porphyrins can be activated by visible light of a specific wavelength (herein 405 nm) to initially generate singlet oxygen or ROS reaction, and consequently kill the target cell. Furthermore, the emission of red fluorescent by porphyrin can be observed during irradiation. (**B**) Schematic picture demonstrates two generations of PICs in the study; Lysine modification leading to heterogeneous IR700-PICs with “average distribution” of DARs 2, 3 and 4 with undefined physical and pharmacokinetic properties. In a click approach, disulfide re-bridging by a pyridazinedione construct yields a homogeneous porphyrin-PIC with a constant DAR4.

**Figure 2 ijms-21-09151-f002:**
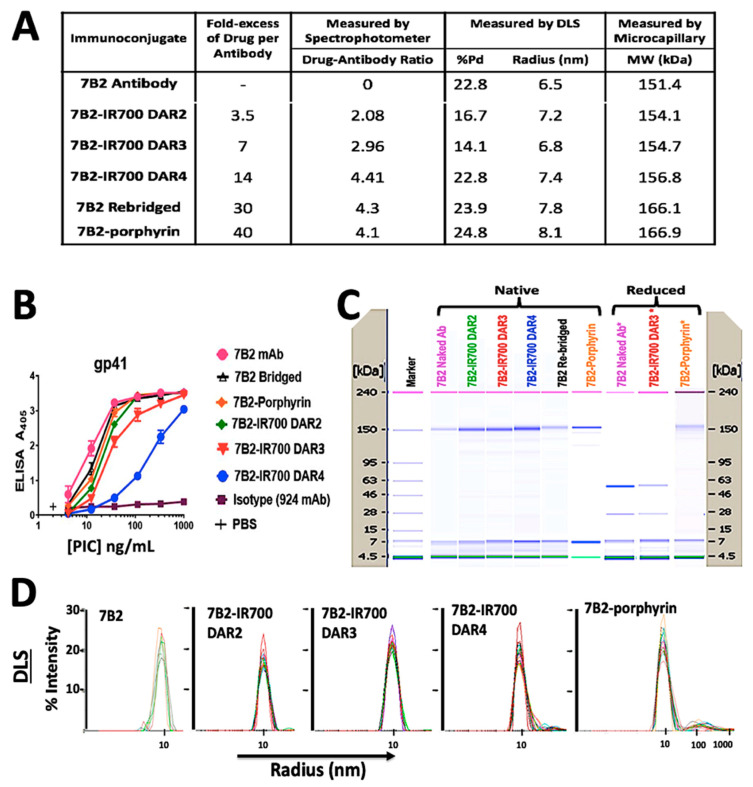
Characterization of photo-immunoconjugates (PICs). (**A**) The table shows products of 7B2 MAb conjugated with different molar concentrations of IR700, rebridged antibody with pyridazinedione and porphyrin. Chemical characterization by UV-Vis spectrophotometry shows the antibody to PS ratio (DAR). In DLS studies, the zeta potential (*ζ*) was estimated according to the Huckel equation. The increase in the molecular weights (MWs) of conjugations, measured by microcapillary electrophoresis, are in confirmation of determined DARs. (**B**) ELISA plates coated with gp41 antigen, as a peptide representing 7B2′s epitope. Results are representative of means of triplicate values with three individual experiments. 924 MAb was used as an isotype control. (**C**) Microcapillary electrophoresis of reduced and non-reduced 7B2 MAb and PICs, before and after irradiation. Unlike 7B2 and 7B2-IR700, the reduced 7B2-porphyrin did not show the separation of light and heavy chains, confirming the accuracy of click chemistry with DAR4. Size standards are indicated on the side of each “gel”. (**D**) DLS Histograms of hydrodynamic radius (R_h_) for 7B2 MAb and PICs, monitoring how the R_h_ preserved during the conjugation process. When IR700 was conjugated with an antibody molecule, ligand dissociation from IR700 affected the shape and solubility of the antibody to which it is attached as well as antibody−antigen complexes.

**Figure 3 ijms-21-09151-f003:**
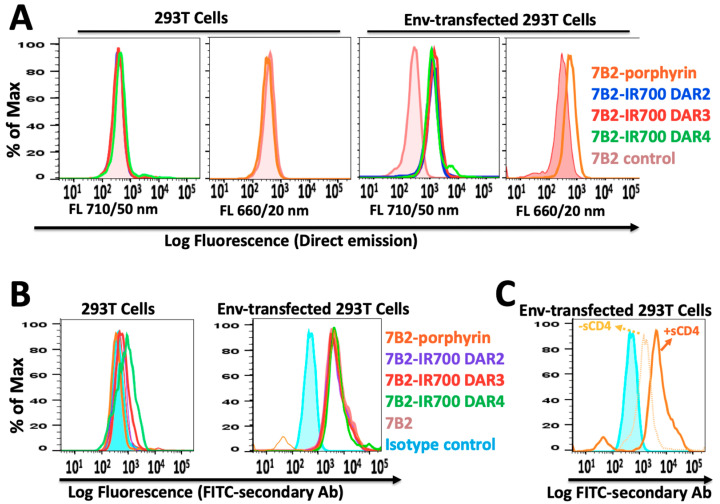
The flow cytometry histogram of PICs by using 293T control cells and Env-transfected 293T cells incubated in PBS + 1% BSA + sodium azide (0.2%). (**A**) PIC binding to 293T cells transfected with 92UG037.8 gp160 (293T/92UG) cells was directly detected through red fluorescent emission from IR700-PICs and porphyrin-PIC, by filter Qdot 705 and filter Qdot 655, respectively. (**B**) By using FITC-secondary immunofluorescence, PICs showed equal binding ability into Env-transfected cells, while 7B2-IR700 DAR4 showed somewhat unspecific binding into the control 293T cells. Results are representative of at least three independent experiments. Isotype control (chimeric RAC18) is shown as blue shaded histogram. (**C**) Studying the binding ability of porphyrin-7B2, in the presence and absence of soluble CD4 (sCD4).

**Figure 4 ijms-21-09151-f004:**
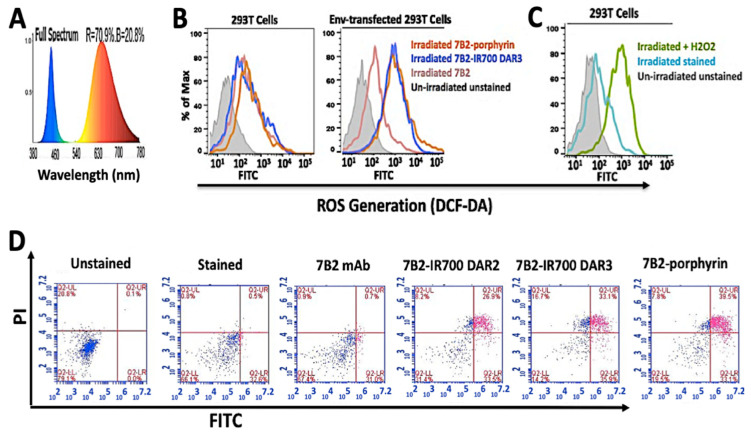
ROS generation and apoptosis assay by PICs. (**A**) The diagram demonstrates a broad spectrum of light (380−780 nm) by a homemade LED device (30 mW/cm^2^) for irradiation of both IR700 and porphyrin PICs, at the same time. Blue spectra can excite the Soret band of porphyrin at 432 nm, meanwhile red spectra excite 7B2-IR700 at 689 nm. (**B**) Samples were kept in the dark (represented by grey) or were irradiated with 50 J/cm^2^. (**C**) As a positive control, 293T cells were treated with 80 µM H2O2 for 24 h in dark. Control group (without H2O2) was incubated under the same conditions. (**D**) In apoptosis assay by Annexin V - FITC/PI, the Env-transfected cells were incubated with PICs for 4 h, then irradiated. PICs showed cell death due to late apoptosis (purple color in subset Upper Right) after 4 h incubation. Control 7B2 antibody showed neither cell death nor apoptosis sign. (*n* = 3).

**Figure 5 ijms-21-09151-f005:**
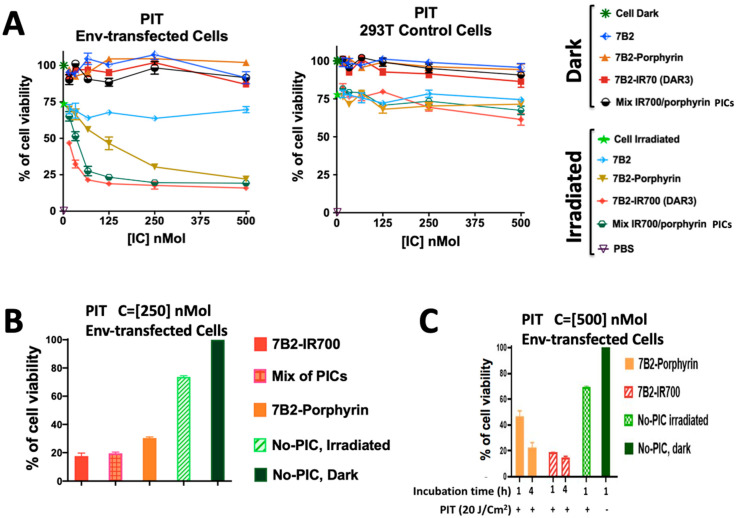
Comparing the photo-cytotoxicity and targeting of PICs by direct cytotoxicity assay. (**A**) Env-transfected 293 T cells (293 T/92UG) and control 293T cells were incubated with 7B2-IR700 (DAR 3), 7B2-porphyrin and an equal molarity mixture of both PICs in the presence of sCD4 (anti-gp120) for 4 h then were assayed by PIT. Data are ± means S.E.M. (*n* = 3) with three individual experiments. Where no error bars are visible, they are obscured by the symbol. (**B**) An equal molarity mixture of porphyrin-PIC with IR700-PIC may increase the cytotoxicity in compare to porphyrin-PIC or the average cytotoxicity of both PICs. Data are ± means S.E.M. (*n* = 3). (**C**) Incubation of transfected cells in two different times (1 and 4 h) before PIT showed the internalization-dependency of Mab-porphyrin to have more photo-cytotoxicity, while the membrane binding of 7B2-IR700 was sufficient to induce cell death. Data are ± means S.E.M. (*n* = 3).

**Figure 6 ijms-21-09151-f006:**
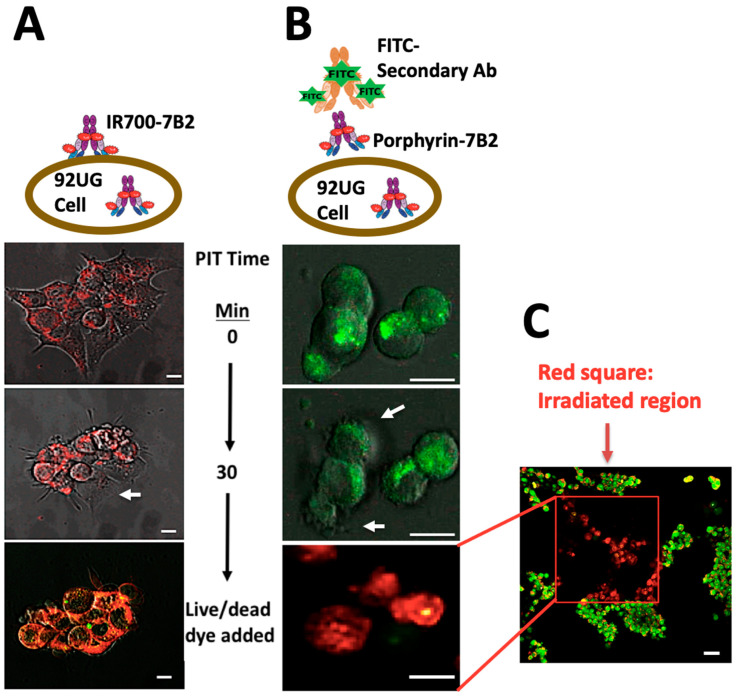
Target-specific cell death in response to PIT in HIV Env-expressing 293T cells. The cells were treated with IR700-7B2 (**A**) or 7B2-porphyrin (**B**) in the presence of sCD4 for 1 h before laser irradiation by microscope. (**A**) The emitted red fluorescent from IR700-7B2 was directly detected. (**B**) 7B2-porphyrin was detected indirectly by using FITC anti-human IgG secondary antibody. After 30 min irradiation, the cell viability was studied by using Acridine Orange and Ethidium Bromide (Live/dead dye). White arrows indicate the bleb formation caused the necrotic cell death. The white bar indicates 10 µm. (**C**) The red square shows the irradiated region that the cells are dead (red color). The non-irradiated region (green color) signifies live cells, confirming no dark toxicity by PIC. The white bar indicates 20 µm.

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
