# Peer review of "Photoimmunotherapy Using Cationic and Anionic Photosensitizer-Antibody Conjugates against HIV Env-Expressing Cells"

_ijms, 2020, doi:10.3390/ijms21239151_

Round 1

Reviewer 1 Report

to the authors:

The authors describe novel methods to weaponize antibodies to bind and clear HIV-1-infected cells, using photosensitizer-conjugation.  There is a strong focus on the biochemistry of the conjugates as compared to the HIV target, which gives a sense that the approach is "plug and play" and could be adapted to a variety of immunotherapeutic targets. The paper is well-written and uses diverse, sophisticated in methods. My only real question is where the work might go from here?  The authors could discuss a little more how the PICs might be used practically. Are they possible to use in preclinical or clinical studies and would the intent be to contribute to possible cure?

A few mostly minor comments are:

I am not sure I understand Fig 4A. This may be because the colors are not distinct. Also is the un-irradiated labeled as black in the key in fact gray in the chart? This confused me initially because the overlapping parts of the blue and brown lines in the chart looks black. The 7B2 and 7B2 porphyrin are very similar colors (brown and a faint purple). Overall,  I would suggest taking a good look at all the colors used here and elsewhere for clarity.

Fig 4C is mentioned in the text before Fig 4A and B. This needs to be fixed so the figure parts are called out in proper order.

Fig 4D. It might be useful to look at the effects of some of these samples on non-transfected cells as a control for non-specific apoptosis. Also, what happens if the cells are not irradiated? (I realize that some of these controls are in fig 5, so I doubt they would be problematic in this assay). Regarding irradiation, I realize this means light activation, but it also brings to mind radioactivity, so I dont know if substituting with "light/photo activation" or another term would avoid that potential misunderstanding for the casual reader.  The sentences  on lines 207 -210 read as being contradictory. What are the differences between early and late stages of apoptosis and how do the data reveal that? Starting the sentence on line 209 with "While.." is not great grammar (though the text is pretty good otherwise). Do the authors mean however?

Figure. 5A and B the text seems to suggest a benefit of the mix, whereas it seems not as good as IR700 alone. I dont see any increase or synergy. I think this needs either explaining or editing. The colors dont entirely match between these figs either and some colors are very similar so this needs fixing for clarity.

Line 268, the authors mention HIV infected cell killing, but as far as I could tell all the data used Env transfected cells.

Line 271. I dont understand how the antibody could be activated after administration to a patient in deep tissue. How would that work? Or am I misunderstanding the text?

Author Response

Dear Referees and Editorial Board Member of International Journal of Molecular Sciences:

Reviewer comments were highly insightful and enabled us to greatly improve the quality of our manuscript. In the following pages, you can see our one-by-one responses to each of the comments of the reviewers. Revisions in the text are shown using yellow highlight, and the rephrased sentences are in red color.

We are looking forward to hearing from you. Thank you very much for your cooperation.

Dr. Francisco Guimaraes

…………………………………………………………………………………………..

Point-by-point response to the reviewer comments:

Reviewer #1:

*The authors describe novel methods to weaponize antibodies to bind and clear HIV-1-infected cells, using photosensitizer-conjugation.  There is a strong focus on the biochemistry of the conjugates as compared to the HIV target, which gives a sense that the approach is "plug and play" and could be adapted to a variety of immunotherapeutic targets. The paper is well-written and uses diverse, sophisticated in methods. My only real question is where the work might go from here?  The authors could discuss a little more how the PICs might be used practically. Are they possible to use in preclinical or clinical studies and would the intent be to contribute to possible cure?

In response to the reviewers 1 and 2: We therefore included a paragraph at page 10, lines 281 to 284.

**A few mostly minor comments are:

*I am not sure I understand Fig 4A. This may be because the colors are not distinct. Also is the un-irradiated labeled as black in the key in fact gray in the chart? This confused me initially because the overlapping parts of the blue and brown lines in the chart looks black. The 7B2 and 7B2 porphyrin are very similar colors (brown and a faint purple). Overall, I would suggest taking a good look at all the colors used here and elsewhere for clarity.

Response: Fig 4 was edited, as suggested by reviewer. We had a look at all the colors for the figures. We changed the brightness of some colors, to make them clear.

*Fig 4C is mentioned in the text before Fig 4A and B. This needs to be fixed so the figure parts are called out in proper order.

Response: Fig 4 was edited, as suggested by reviewer. The numbering of Fig 4 (A-D) was edited and sorted out in proper order, in page 6, section 2.3. ROS generation.

*Fig 4D. It might be useful to look at the effects of some of these samples on non-transfected cells as a control for non-specific apoptosis. Also, what happens if the cells are not irradiated? (I realize that some of these controls are in fig 5, so I doubt they would be problematic in this assay). Regarding irradiation, I realize this means light activation, but it also brings to mind radioactivity, so I dont know if substituting with "light/photo activation" or another term would avoid that potential misunderstanding for the casual reader.  The sentences  on lines 207 -210 read as being contradictory. What are the differences between early and late stages of apoptosis and how do the data reveal that? Starting the sentence on line 209 with "While.." is not great grammar (though the text is pretty good otherwise). Do the authors mean however?

Response: In Fig 4, the controls include the unstained, stained and 7B2-incubated cells. As expected, the non-irradiated cells did not show features of apoptosis, therefore data not shown here. In the diagrams, purple color represents the apoptosis signs. No features related to early stages in the lower right (purple color in subset Q-LR) square was observed in response to PICs.

 Changes: In Fig 4D, this term added in line 205: "(purple color in subset Upper Right)".

In line 211, this term added: "(Figure 4D, purple color in subset Lower Right)".

In line 211, the word "While" deleted.

In line 214, this sentence added: "the non-irradiated cells did not show sign of apoptosis (data not shown)".

*Figure. 5A and B the text seems to suggest a benefit of the mix, whereas it seems not as good as IR700 alone. I dont see any increase or synergy. I think this needs either explaining or editing. The colors dont entirely match between these figs either and some colors are very similar so this needs fixing for clarity.

Response and changes: The synergic effect is more significant in the concentration of 250 nMol. Hence, we replaced the graph in Fig 5B with a graph demonstrating the cell viability in the concentration of 250 nMol. In lines 240-243, the effect of this combination has explained. Furthermore, as suggested, figure 5 edited, regarding the clarity.

*Line 268, the authors mention HIV infected cell killing, but as far as I could tell all the data used Env transfected cells. Response and changes: The term replaced with "HIV-1 Env-expressing cells" in the line 270.

*Line 271. I dont understand how the antibody could be activated after administration to a patient in deep tissue. How would that work? Or am I misunderstanding the text?

Response: Regarding this question, we added more explanations at page 10, lines 275 to 280 as follows:

Considering the fundamental restriction of light penetration, we suggest optimizing the excitation of PSs in deep tissue that may give us a highly flexible theranostic platform for HIV researches. This inevitably issue dictates the use of high-power irradiation (43). The other alternatives for light delivery to the PICs in deep tissue could be the application of cellular-scale inorganic light-emitting diode (µ-ILED) arrays (44), or upconversion nanoparticles (UCNP) in the field of PIT (45).

Reviewer 2 Report

The manuscript „Photoimmunotherapy using cationic and anionic photosensitizer-antibody conjugates against HIV Env-expressing cells” by Sadraeian et al. describe the initial validation of targeted killing of HIV-1 Env-expressing cells using an approach that combines photodynamic therapy and immunotherapy. Specifically, an anti-gp41 antibody (7B2) is conjugated to two individual photosensitizers. These photoimmunoconjugates, upon irradiation, induce targeted cytotoxicity of cells heterologously expressing HIV-1 Env at the cell surface. While this report is interesting and appears to propose a novel approach that might be useful especially in the context of HIV-1 cure, a more detailed validation would have been beneficial.

Major issues:

The authors do not describe the context in which they propose to use this approach, when further developed, in vivo. It is unclear how the levels of surface HIV-1 Env which are expressed in the tested system relate to the levels that are expressed on the surface of HIV-1-infected cells, especially in the context of HIV-1 latency. It would be very useful if the authors tested this experimentally, e.g. by testing their killing approach on HIV-1 (latently) infected cells, or by conducting a comparison of HIV-1 Env surface levels in infected and transfected cells. At least, this issue and the implication of potential differences need to be discussed.

The authors do not discuss the prospects of this approach in an in vivo set-up. Are the required doses of irradiation feasible for an in vivo treatment? How does the irradiation dose compared to doses applied in patients undergoing cancer therapy?

The detection of cell death by addition of the live-dead dyes orange acrylamide and ethidium bromide is a bit inconclusive. By which fluorescence change are live/dead cells identified? What is the morphology of healthy, living cells after addition of the live/dead dye?

In the Materials & Methods part, a cell proliferation assay is described, but no data using this method is shown.

The link to the supplemental data seems to be broken, and the files in the zip folder containing the supplementary data could not be opened for unknown reasons.

Minor issues:

Line 22: two photosensitizers: the abbreviation for photosensitizers (PS) should be introduced here.

Figure 2DF seems to be not discussed in the text.

Figure 5B: the information about the numbers of experiments and the standard deviation is missing.

Line 245: “red fluorescent emission” should be changed to “red fluorescence emission”

Line 266 “cell death in HIV-transfected cells” should be changed to “cell death in HIV Env-transfected cells”

Line 269: “directly killing HIV infected-cell” should be changed to “directly killing HIV-1 Env-expressing cells”

Line 282: “293T cells were used as uninfected control cells” should be changed to “293T cells were used as untransfected control cells”

Author Response

Dear Referees and Editorial Board Member of International Journal of Molecular Sciences:

Reviewer comments were highly insightful and enabled us to greatly improve the quality of our manuscript. In the following pages, you can see our one-by-one responses to each of the comments of the reviewers. Revisions in the text are shown using yellow highlight, and the rephrased sentences are in red color.

We are looking forward to hearing from you. Thank you very much for your cooperation.

Dr. Francisco Guimaraes

…………………………………………………………………………………………..

Point-by-point response to the reviewer comments:

Reviewers #2 and #3:

**The manuscript „Photoimmunotherapy using cationic and anionic photosensitizer-antibody conjugates against HIV Env-expressing cells” by Sadraeian et al. describe the initial validation of targeted killing of HIV-1 Env-expressing cells using an approach that combines photodynamic therapy and immunotherapy. Specifically, an anti-gp41 antibody (7B2) is conjugated to two individual photosensitizers. These photoimmunoconjugates, upon irradiation, induce targeted cytotoxicity of cells heterologously expressing HIV-1 Env at the cell surface. While this report is interesting and appears to propose a novel approach that might be useful especially in the context of HIV-1 cure, a more detailed validation would have been beneficial.

**Major issues:

Point 1. The reviewer mentions that “The authors do not describe the context in which they propose to use this approach, when further developed, in vivo. It is unclear how the levels of surface HIV-1 Env which are expressed in the tested system relate to the levels that are expressed on the surface of HIV-1-infected cells, especially in the context of HIV-1 latency. It would be very useful if the authors tested this experimentally, e.g. by testing their killing approach on HIV-1 (latently) infected cells, or by conducting a comparison of HIV-1 Env surface levels in infected and transfected cells.”  Reviewer also suggested that “At least, this issue and the implication of potential differences need to be discussed.”

Response: We therefore included the following paragraph at page 10, lines 281 to 289 as follows. "Approaches that focus on the combination of this PIT strategy with “shock and kill” technique, could be a cornerstone of future research efforts towards an HIV cure (46). The photo-cytotoxicity on the cell membrane triggered by the presence of HIV proteins can be an interesting to be proven strategy aiming the elimination of HIV latent reservoir in vivo. Previously, we observed different levels of surface HIV Env on the transfected and infected cells which may reflect differences due to laboratory-adapted versus clinical isolates (11). However, it is still unclear how these discrepancies may relate to the expressed levels of HIV Env on the surface of HIV-infected cells, especially in the context of HIV-1 latency. We therefore recognize that further experimental killing approach on HIV-1 latently infected cells would add fundamental knowledge to the proposed PIT strategy."

Point 2. The reviewer mentions that "The authors do not discuss the prospects of this approach in an in vivo set-up. Are the required doses of irradiation feasible for an in vivo treatment? How does the irradiation dose compared to doses applied in patients undergoing cancer therapy?"

Response: The doses of irradiation in this in vitro study is comparable to our recent in vivo study feasible for cancer therapy. We therefore included the following paragraph at page 7, lines 222 to 224 as follows: "As a prospect of this approach in an in vivo study, the doses of irradiation can be increased to maximum energy fluence of 100 J/ cm2, depending on the selected wavelength, however, visible light irradiation is unlikely to be toxic (43)."

Point 3. The reviewer mentions that "The detection of cell death by addition of the live-dead dyes orange acrylamide and ethidium bromide is a bit inconclusive. By which fluorescence change are live/dead cells identified? What is the morphology of healthy, living cells after addition of the live/dead dye?"

Response: After irradiation, the occurrence of cell death (irradiated region, red fluorescence emission) was compared with live cells (non-irradiated region, green fluorescence emission) by double-staining Acridine Orange-Ethidium Bromide (AO/EB). In another experiment, live-dead cells were studies using propidium iodide (PI) (Microscopic videos, in the supplementary methods). Detailed conditions and settings are described in the supplementary methods. Furthermore, propidium iodide (PI) and Annexin V-FITC staining were used for Apoptosis / cell death assay (Figure 4D). The morphological changes were observed by optical microscopy before and after irradiation (Figures, in the supplementary methods).

Changes in the manuscript: Based on the Reviewer’s comment, the term "(AO/EB)" added in lines 556 and 265. The description of AO/EB Staining added in; lines 415 to 416.

Point 4. In the Materials & Methods part, a cell proliferation assay is described, but no data using this method is shown.

Response: Data related to cell proliferation assay are shown in Figure 5 and Supplementary Figure S2. Changes: In the Materials & Methods part, we changed the term " cell proliferation assay " to " Photo-cytotoxicity Assay " that is more general, in line 392.

Point 5. The link to the supplemental data seems to be broken, and the files in the zip folder containing the supplementary data could not be opened for unknown reasons.

Response: We checked the link to the supplemental data, and we did not find any problem. Briefly, we opened the zip folder, then opened two microscopic videos by VLC software, and the pdf file with acrobat reader. As an alternative, you may find the supplemental data via the link to the "Data availability'. Furthermore, we will ask the assigned editor (Tina Hong) to have a double check it.

**Minor issues:

Line 22: two photosensitizers: the abbreviation for photosensitizers (PS) should be introduced here. Response and changes: Term "(PSs)" added in lines 22.

Figure 2D seems to be not discussed in the text. Response: Figure 2D is discussed in the lines 112-114.

Figure 5B: the information about the numbers of experiments and the standard deviation is missing. Response: The information added in the line 235.

Line 245: “red fluorescent emission” should be changed to “red fluorescence emission”. Response: The term edited in the line 250.

Line 266 “cell death in HIV-transfected cells” should be changed to “cell death in HIV Env-transfected cells”. Response: The term edited in the line 270.

Line 269: “directly killing HIV infected-cell” should be changed to “directly killing HIV-1 Env-expressing cells”. Response: The term edited in the line 273.

Line 282: “293T cells were used as uninfected control cells” should be changed to “293T cells were used as untransfected control cells”. Response: The term edited in the line 301.

Reviewer 3 Report

The manuscript „Photoimmunotherapy using cationic and anionic photosensitizer-antibody conjugates against HIV Env-expressing cells” by Sadraeian et al. describe the initial validation of targeted killing of HIV-1 Env-expressing cells using an approach that combines photodynamic therapy and immunotherapy. Specifically, an anti-gp41 antibody (7B2) is conjugated to two individual photosensitizers. These photoimmunoconjugates, upon irradiation, induce targeted cytotoxicity of cells heterologously expressing HIV-1 Env at the cell surface. While this report is interesting and appears to propose a novel approach that might be useful especially in the context of HIV-1 cure, a more detailed validation would have been beneficial.

Major issues:

The authors do not describe the context in which they propose to use this approach, when further developed, in vivo. It is unclear how the levels of surface HIV-1 Env which are expressed in the tested system relate to the levels that are expressed on the surface of HIV-1-infected cells, especially in the context of HIV-1 latency. It would be very useful if the authors tested this experimentally, e.g. by testing their killing approach on HIV-1 (latently) infected cells, or by conducting a comparison of HIV-1 Env surface levels in infected and transfected cells. At least, this issue and the implication of potential differences need to be discussed.

The authors do not discuss the prospects of this approach in an in vivo set-up. Are the required doses of irradiation feasible for an in vivo treatment? How does the irradiation dose compared to doses applied in patients undergoing cancer therapy?

The detection of cell death by addition of the live-dead dyes orange acrylamide and ethidium bromide is a bit inconclusive. By which fluorescence change are live/dead cells identified? What is the morphology of healthy, living cells after addition of the live/dead dye?

In the Materials & Methods part, a cell proliferation assay is described, but no data using this method is shown.

The link to the supplemental data seems to be broken, and the files in the zip folder containing the supplementary data could not be opened for unknown reasons.

Minor issues:

Line 22: two photosensitizers: the abbreviation for photosensitizers (PS) should be introduced here.

Figure 2DF seems to be not discussed in the text.

Figure 5B: the information about the numbers of experiments and the standard deviation is missing.

Line 245: “red fluorescent emission” should be changed to “red fluorescence emission”

Line 266 “cell death in HIV-transfected cells” should be changed to “cell death in HIV Env-transfected cells”

Line 269: “directly killing HIV infected-cell” should be changed to “directly killing HIV-1 Env-expressing cells”

Line 282: “293T cells were used as uninfected control cells” should be changed to “293T cells were used as untransfected control cells”

Author Response

(The authors gave the same response as above.)
